# Long-term neuropsychiatric and neuropsychological impact of the pandemic in Italian COVID-19 family clusters, including children and parents

Costanza Di Chiara[1,2]*, Martina Ferrarese[3], Riccardo Boracchini[4], Anna Cantarutti[4], Anna Letizia Tibaldo[3], Chiara Stefanni[3], Daniele Donà[1,2], Marica De Pieri[1], Alessia Raffagnato[3], Benedetta Tascini[3], Marina Miscioscia[3], Federica Occhino[3], Carlo Giaquinto[1,2], Michela Gatta[3]

**1** Department for Women's and Child's Health, University Hospital of Padova, Padua, Italy, **2** Penta-Child Health Research, Padua, Italy, **3** Child and Adolescent Neuropsychiatric Unit, Department of Women's and Children's Health, University Hospital of Padua, Padua, Italy, **4** Department of Statistics and Quantitative Methods, Division of Biostatistics, Epidemiology and Public Health, Laboratory of Healthcare Research and Pharmacoepidemiology, University of Milano-Bicocca, Milan, Italy

☯ These authors contributed equally to this work
* costanza.dichiara@phd.unipd.it

## Abstract

### Aim

This study investigated the long-term neuropsychiatric and neuropsychological impact of COVID-19 on children and their parents in households with COVID-19 exposure.

### Methods

A prospective cohort study was conducted on 46 families attending the COVID-19 Follow-up Clinic at the Department for Women's and Children's Health, Padua (Italy) from December 2021 to November 2022. Self-perceived stress-related, emotional-behavioral, and post-traumatic stress (PTSD-related) symptoms were assessed in both children and parents. Children with underlying neuropsychiatric conditions were excluded from the study.

### Results

A total of 81 parents (median age = 38 years [IQR: 43–48], females = 44 [54.3%]), and 77 children (median age = 8 years [IQR: 5–11], females = 33 [42.9%]) participated in the study. Overall,125 (79%) and 33 (21%) participants were classified as COVID-19 cases and non-COVID-19 cases, respectively. The time interval between the COVID-19 family outbreak and the neuropsychiatric and psychological assessment was ≤4 months (median=3 months [IQR=0]) for 89 (56.3%) participants and >4 months for 69 (43.8%) (median=11.5 months [IQR=5–12]) participants. A total of 136 (86.1%) participants reported stress-related symptoms, with emotional stress being the most common.

**Data availability statement:** Minimal data is available upon request in line with the ethical considerations and requirements set forth by the authors' institutional review board and with the informed consent that has been signed by participants/caregivers. The data were collected and stored in a pseudo-anonymized format at the Department for Child and Women's Health of the University of Padova, which is the corresponding author's affiliation. The clinical data of the current manuscript are available from the corresponding author (Dr. Costanza Di Chiara - costanza.dichiara@phd.unipd.it) and the local Ethics Committee, Operational Unit for Projects and Clinical Research, University Hospital of Padova, Italy (prc.unitaricercaclinica@aopd.veneto.it).

**Funding:** The author(s) received no specific funding for this work.

**Competing interests:** The authors have declared that no competing interests exist.

A positive correlation was observed between self-perceived stress-related symptoms in children and their parents within the same family (r=0.53, p=0.0005). Among children aged 6–18 years, 16 (37.2%) had clinical scores for internalizing symptoms at the Child Behavior Checklist (CBCL), while none children aged 1.5–5 years showed clinical score for internalizing symptoms. Similarly, total difficulty scores at the Strengths and Difficulties Questionnaire (SDQ 4–17) and assessment of PTSD-related symptoms through the Trauma Symptom Checklist for Young Children (TSCYC) questionnaire were within non-clinical cut-offs in 45 (84.9%) and 43 (75.4%) children aged 3–12 years, respectively. The Trauma Symptom Checklist for Children (TSCC) resulted in the non-clinical cut-off for 36 (92.3%) children aged 8–18 years. While a higher prevalence of self-perceived stress-related symptoms was found in COVID-19 cases compared to non-COVID-19 cases (p=.01), no differences were observed for emotional-behavioral and PTSD-related symptoms between the two groups.

## Conclusions

This study documented the impact of the COVID-19 pandemic on Italian children and their family's stress levels. Further research is needed to confirm our findings and explore the long-term effects of the pandemic on families.

## 1. Introduction

The consequences of the COVID-19 pandemic continue to affect human health and mental well-being worldwide. Beyond the acute phase of the illness, a consistent number of patients, including children and adults, have developed post-acute sequelae of SARS-CoV-2 infection, known as long COVID.[1,2] Long COVID has proven to be a perplexing condition, characterized by a broad spectrum of symptoms that persist for extended periods, ranging from weeks to months and sometimes even years following the initial infection.[3,4] Among the diverse spectrum of symptoms, neurological and psychological manifestations have emerged as prominent features of long COVID in both pediatric and adult populations.[1,5] Similar to adults, children enduring the aftermath of long COVID have been documented to experience various symptoms, spanning from fatigue and headache to anxiety, insomnia, memory loss, and cognitive impairment, all of which potentially impact the quality of patients' lives.[6–8] In addition, emerging evidence underscored the persistence of neuropsychiatric symptoms in individuals more than 12 months after SARS-CoV-2 infection,[9] although their long-term trajectory may vary between children and adults.[10]

However, a knowledge gap persists regarding the prevalence and nature of neuropsychiatric and neuropsychological sequelae following COVID-19 in children and adolescents. While several studies have documented the neuropsychiatric and/or neuropsychological sequelae of COVID-19 in the pediatric population, most of them were based on online surveys or parental reports,[11] while studies including comprehensive neuropsychological assessments in children are still limited.

Moreover, while it is currently hypothesized that the variety of neurological and psychiatric sequelae post-COVID-19 may have a multifactorial pathogenesis, spanning from neuronal and blood vessel injury to neuroinflammation and brain stem hypometabolism, it is also assumed that most of the manifestations may be a consequence of social isolation rather than direct effects of SARS-CoV-2.[12–15] In line with this, an increased risk of post-traumatic

stress disorder (PTSD) post-COVID-19 has been largely observed, showing a positive correlation with mental health issues in the pediatric population.[16,17] Thus, there exists a need for studies that delve into the comprehensive neuropsychiatric evaluations of children following COVID-19.

The family environment had also been affected, playing a crucial role in the well-being of children.[18,19] The social restrictions imposed during the pandemic have notably disrupted daily family routines, increasing stress in children and parents.[20] This stress potentially leads to the deterioration of children's mental health as well as exacerbates emotional and behavioral symptoms in adolescents with underlying neuropsychiatric disorders.[21] Considering the significant influence of family mental health on children's well-being, it has become imperative to delve deeper into the psychosocial impact of the COVID-19 pandemic on children and their parents.

This study aimed to assess the prevalence and characteristics of neuropsychiatric symptoms and neuropsychological sequelae following COVID-19 in a cohort of Italian COVID-19 family clusters.

## 2. Materials and methods

### 2.1 Study design, setting, and participants

We conducted a single-centre, prospective cohort study to explore the neuropsychiatric and neuropsychological impact of COVID-19 infection and the pandemic on Italian families, including children, older siblings, and their parents.

The study was conducted on families who attended the COVID-19 Family Cluster Follow-up Clinic (CovFC) at the Department of Women's and Children's Health, University Hospital of Padua (Veneto region, Italy). The CovFC is a multidisciplinary pediatric outpatient clinic, staffed by pediatricians with expertise in pediatric infectious diseases, cardiology, pneumology, psychology, neuropsychiatry, virologists, and immunologists. It aims to provide comprehensive clinical and immunological follow-up to families who have experienced COVID-19 within their households. The features of the CovFC have been previously described elsewhere.[22] Families with at least one child aged 0–14 years and at least one family member with a previous laboratory-confirmed diagnosis of COVID-19 were enrolled in the CovFC at least 8–12 weeks after infection.

The neuropsychiatric assessment was part of the clinical follow-up conducted at the CovFC.

At enrollment between December 17, 2021, and November 3, 2022, families were informed of the opportunity to participate in a neuropsychiatric and neuropsychological assessment. This assessment was offered to children and older siblings aged 0–18 years, referred to as "children" hereafter, as well as their parents, and was conducted by the team of neuropsychiatrists affiliated with the CovFC network. Children who had received a neuropsychiatric diagnosis prior to the date of the SARS-CoV-2 infection were excluded. The study protocol was approved by the local Ethics Committee (Prot. N° 0070714 of 24th November 2020; last amendment Prot. N° 0024018 of 5th April 2022). Parents or legally authorized representatives were informed of the research proposal and provided written consent to use the routine patient-based data for research purposes.

### 2.2 Assessments

For each family member, sociodemographic (i.e., age, sex, and socio-economic status [SES] [23]) and clinical data were collected, including past medical history (i.e., non-neuropsychiatric underlying conditions and familiarity with neuropsychiatric conditions),

COVID-19-related factors (i.e., acute and post-acute symptoms of COVID-19, variation of biological functions [(i.e., sleep-wake rhythm, appetite, and body weight)] and activities [(i.e., educational/working routines, physical activity, and daily habits)] during home isolation), and nasopharyngeal swab [NPS] date(s) and result(s)). Additionally, a blood sample was collected from all family members to detect anti-SARS-CoV-2-specific antibodies.[24,25]

Data were anonymized and entered into a web-based database using the REDCap® platform (Vanderbilt University, Tennessee).

The neuropsychiatric evaluation aimed to assess the presence of neuropsychiatric symptoms and neurocognitive impairment following experiencing the COVID-19 family outbreak and included the following evaluations, which are summarized in Table S1:

a) The assessment of self-perceived stress-related symptoms through a self-administered questionnaire was completed by parents and children aged ≥10 years, while, for children aged <10 years, the questionnaire was filled in by one of the parents. The questionnaire collected a broad spectrum of stress-related symptoms divided into four different categories: 1) physical, 2) behavioural, 3) emotional, and 4) cognitive symptoms. The entire questionnaire has been previously described,[26] and provided as **Supplementary Questionnaire**.

b) The Child Behavior Checklist (CBCL), Italian standardized version for children aged 1.5–5 and another one for 6–18 years, was completed by the parent (mother or father) and assessed emotional-behavioural problems.[27] Cut-off for clinical significance was set at >98%ile, while score ≤95–98%ile was considered normal-borderline.

c) The Strengths and Difficulties Questionnaire (SDQ 4–17), Italian standardized version for children aged 3–12 years, is a questionnaire completed by a parent (mother or father) and assesses emotional-behavioural problems, including problems in conduct, attention/hyperactivity, and relationships with peers.[28] Cut-off for clinical significance was set at >90%ile; while scores of 80–90%ile, and <80%ile were considered subclinical and normal, respectively. Clinical and subclinical cut-offs for each scale are described in Table S2.

d) The Trauma Symptom Checklist for Young Children (TSCYC) and the Trauma Symptom Checklist for Children (TSCC), Italian standardized versions for children aged 3–12 and 8–18 years, respectively.[29] They are questionnaires filled out by parents, evaluating the presence of PTSD-related symptoms. Clinical and subclinical cut-offs for each subscale are described in Table S3.

e) The Leiter International Performance Scale, 3rd edition (Leiter-3), Italian standardized version for children aged 3–18 years, evaluating neurocognitive outcomes (i.e., attention, memory, and cognitive interference).[30] The total scores of "Non-Verbal Memory" and "Processing Speed" were considered as "below average" if <89 corresponding to <24%ile, "average" if between 90–109 corresponding to 25–75%ile, and "above average" if ≥110 corresponding to >75%ile.

f) The Depression Anxiety Stress Scale 21 (DASS-21), Italian standardized version, which is a psychometric test for adults and assesses the presence of depression, anxiety and stress.[31] Cut-offs were considered as clinical if: depression ≥9, anxiety ≥7, and stress ≥14.

g) The Impact of Event Scale Revised (IES-R), Italian standardized version, which is a 22-element self-administered questionnaire completed by both parents and assesses their self-perceived PTSD-related symptoms, estimating the perceived discomfort caused by the traumatic event.[32] Cut-off for clinical significance was set at ≥1.1.

## 2.3 Definitions

Participants were categorized as "COVID-19 cases" if they exhibited virological positivity for SARS-CoV-2, which was confirmed either through real-time RT-PCR or an antigenic test officially conducted by the regional government at the onset of infection. Alternatively, for individuals who were not vaccinated, confirmation was based on the detection of anti-SARS-CoV-2 antibodies through one of the two tests performed at the time of enrollment in the CovFC. Participants who lacked analytical evidence of SARS-CoV-2 infection were designated as "non-COVID-19" cases.

For each COVID-19 family cluster, a *baseline* date was established, defined as the earliest date among the infection-onset dates (the date of the first positive NPS) for all family members. To better evaluate the short-medium and long-term impact of COVID-19 infection and/ or isolation, the timeframe between the *baseline* date and the family neuropsychiatric assessment was divided into ≤4 months and >4 months.

The severity of COVID-19 was categorized as mild, moderate, severe, or critical in each case, following the classification provided by the World Health Organization (WHO).[33]

COVID-19 cases were further categorized based on the predominant circulating SARS-CoV-2 Variant of Concern (VOC) in the Veneto Region at the time of the COVID-19 family cluster's baseline date. This classification was determined using the CovSPECTRUM platform, which relies on surveillance data.[34] The primary VOCs considered for analysis included Parental, Delta, and Omicron (comprising B.1.1.529, BA.2, BA.4, and BA.5 sublineages). Specifically, any SARS-CoV-2 infection occurring in the Veneto region from February 2020 to June 29, 2021, was attributed a probability greater than 50% of being caused by the Parental VOC. In contrast, infections occurring from June 30, 2021, to December 25, 2021, were attributed a probability greater than 50% of being caused by the Delta VOC. Lastly, infections from December 26, 2021, to November 23, 2022, had a probability greater than 96% of being associated with the Omicron VOC.[34]

## 2.4 Statistical analysis

Sociodemographic and clinical characteristics were appropriately categorized and summarized through frequency and percentage and median and interquartile range (IQR) to represent categorical and continuous variables, respectively. Chi-squared test, Fisher exact test and Wilcoxon's test were used to evaluate differences between CovFC enrolled members as appropriate.

Considering as unit statistic of interest the family cluster, we evaluated the correlation between (i) the number of neuropsychiatric and neuropsychological sequelae in children and their parents; (ii) the mean TSCC/TSCY/IES scores evaluated among children and parents, respectively, as well as the mean score among daughters/sons and mother/father of each family cluster; and (iii) the mean score of CBCL and DASS. The Pearson's and Spearman's correlations were used appropriately.

A significant two-sided P-value was considered if <0.05. All statistical analyzes were performed with SAS software (version 9.4, SAS Institute, Cary, NC, USA).

## 3. Results

### 3.1 Study population's characteristics

A total of 46 families participated in the study, comprising 81 parents (median age = 38 years [IQR: 43–48], females = 44 [54.3%]), and 77 children (median age = 8 years [IQR: 5–11], females = 33 [42.9%]). Eleven families (23.9%) included a single parent, while twenty-three

(50%), nineteen (41.3%), and four (8.7%) families had two, one, and three children, respectively. Sociodemographic characteristics of the study population are presented in Table 1.

Out of 81 parents and 77 children, 21 (25.9%) and 36 (46.8%), respectively, had a family history of neuropsychiatric disorders.

Out of 158 patients in the cohort, 125 (79%) participants were classified as COVID-19 cases, while 33 (21%) were categorized as non-COVID-19 cases. Among them, 40 (25.3%), 64 (40.5%), and 54 (34.2%) experienced their COVID-19 family outbreak during the Parental, Delta, and Omicron waves, respectively. Of the 77 children, 65 (84.4%) were COVID-19 cases, with 8 (12.3%) experiencing asymptomatic and 57 (87.7%) mild COVID-19; none required hospitalization for moderate/severe infection. Among the 81 parents, 60 (74.1%) were COVID-19 cases, with 4 (6.7%) having asymptomatic and 53 (88.3%) mild COVID-19, while 3 (5%) parents experienced moderate/severe infection characterized by radiologically confirmed pneumonia requiring hospitalization. No participants reported short- or long-term sequelae following COVID-19.

**Table 1. Sociodemographic and COVID-19-related characteristics of children and parents enrolled in the study.**

|  | Overall (N = 158) | | Children (N = 77) | | Parents (N = 81) | |
|---|---|---|---|---|---|---|
|  | Non COVID-19 cases | COVID-19 cases | Non COVID-19 cases | COVID-19 cases | Non COVID-19 cases | COVID-19 cases |
| N (%) | 33 (21) | 125 (79) | 12 (15.6) | 65 (84.4) | 21 (25.9) | 60 (74.1) |
| Sex, F, (%) | 13 (39.4) | 64 (51.2) | 3 (25) | 30 (46.2) | 10 (47.6) | 34 (56.7) |
| Age, median (IQR) | – | – | 6.5 (4–12) | 8 (5–11) | 43 (40–47) | 43 (38–48) |
| SES[a], median (P25 - P75) | 45 (27.5 - 53) | 42 (29 - 52.5) | 45 (24.5 - 52.5) | 40.8 (31.3 - 51) | 44.5 (30 - 55.5) | 42 (27 - 55) |
| Familiarity for neuropsychiatric conditions, N (%) (1 missing value) | 9 (27.3) | 48 (38.7) | 5 (41.7) | 31 (48.4) | 4 (19.1) | 17 (28.3) |
| Non-neuropsychiatric underlying conditions, N (%) | 9 (26.5) | 41 (33.1) | 3 (23.1) | 20 (31.3) | 6 (28.6) | 21 (35) |
| COVID-19 clinical classification[b], N (%) |  |  |  |  |  |  |
| Asymptomatic | – | 12 (9.6) | – | 8 (12.3) | – | 4 (6.7) |
| Mild[c] | – | 110 (88) | – | 57 (87.7) | – | 53 (88.3) |
| Moderate/severe | – | 3 (2.4) | – | 0 (0) | – | 3 (5) |
| COVID-19 family cluster's pandemic wave, N (%) |  |  |  |  |  |  |
| Parental | 8 (24.2) | 32 (25.6) | 5 (41.67) | 14 (21.5) | 3 (14.3) | 18 (30) |
| Delta | 17 (51.5) | 47 (37.6) | 5 (41.67) | 27 (41.5) | 12 (57.1) | 20 (33.3) |
| Omicron | 8 (24.2) | 46 (36.8) | 2 (16.66) | 24 (37) | 6 (28.6) | 22 (36.7) |
| Length of home isolation, N (%) |  |  |  |  |  |  |
| <3 weeks | 14 (42.4) | 48 (38.4) | 4 (33.3) | 27 (41.5) | 10 (47.6) | 21 (35) |
| ≥3 weeks | 19 (57.6) | 77 (61.6) | 8 (66.7) | 38 (58.5) | 11 (52.4) | 39 (65) |
| Time between baseline and NPI visit, median (P25 - P75) | 130 (95–251) | 113 (89–299) | 164 (123–369.5) | 107 (86–219) | 115 (90–157) | 125.5 (92.5–364) |
| Variation of biologic functions[d] during home isolation, N (%) | 14 (42.4) | 61 (48.8) | 6 (50) | 32 (49.2) | 8 (38.1) | 29 (48.3) |
| Variation of activities[e] during home isolation | 8 (24.2) | 43 (34.4) | 7 (58.3) | 36 (55.4) | 1 (4.8) | 7 (7.8) |

[a]SES = Socio-economic status;

[b]World Health Organization (WHO) COVID-19 clinical classification;

[c]Among children, the most common symptoms were fever (64.9%), rhinitis (35.1%), headache (24.6%), asthenia (17.5%), and cough (15.8%).

[d]Variation of biologic functions was defined as at least one variation among sleep-wake rhythm, appetite, and body weight;

[e]Variation of activities was defined as at least one variation among educational/working routines, physical activity, and daily habits.

The time between the baseline date and the neuropsychological assessment was ≤4 months for 89 (56.3%) participants and >4 months for 69 (43.8%) participants. Specifically, the median time between the baseline and the neuropsychological assessment was 3 (IQR=3–3) for those assessed within ≤4 months, and 11.5 (IQR=5–12) months for those assessed after >4 months.

### 3.2  Self-perceived stress-related symptoms among children and parents

Table 2 (Table S4) presents the self-perceived stress-related symptoms among the participants. Overall, 136 (86.1%) participants reported experiencing stress-related symptoms, with emotional stress-related symptoms being the most common (N=112, 70.9%), followed by physical (N=87, 55.1%), cognitive (N=73, 46.2%), and behavioral (N=48, 30.4%) stress-related symptoms.

No significant differences were observed in the number of self-perceived symptoms between children and parents. The presence of at least one non-neuropsychiatric underlying condition was associated with a higher number of physical (p<.01), behavioural (p<.01), and emotional (p<.01) stress-related symptoms. Similarly, familial neuropsychiatric conditions were associated with a higher number of emotional stress-related symptoms (p=.01).

A higher number of physical stress-related symptoms was noted in COVID-19 compared to non-COVID-19 cases (p=.01). Interestingly, COVID-19 cases that experienced symptomatic COVID-19 reported a higher number of physical (p<.01), emotional (p<.01), and cognitive (p=.01) stress-related symptoms compared to those with asymptomatic COVID-19.

Furthermore, physical (p<.01), behavioural (p=.01), emotional (p<.01), and cognitive (p=.04) stress-related symptoms were more common among participants who experienced their COVID-19 family outbreak during the Delta and Omicron waves. Moreover, behavioural (p=.05), emotional (p<.01), and cognitive (p=.02) stress-related symptoms were more common in participants who underwent a longer home isolation period (≥3 weeks) compared to those with a shorter isolation period (<3 weeks).

### 3.3  *Neuropsychiatric and neuropsychological assessment of children*

In the neuropsychiatric and neuropsychological assessment of children, emotional-behavioural and PTSD-related symptoms were evaluated using the CBCL, SDQ-4–17, TSCYC, and TSCC questionnaires. Parents completed the CBCL questionnaire for 22/24 (91.7%) and 47/53 (88.7%) children aged 1.5–5 and 6–18 years, respectively.

Fig 1 shows an overall discordance between the self-perceived stress-related symptoms and children's neuropsychiatric and neuropsychological assessment through the considered questionnaires. Panel A reports the prevalence of children with self-perceived stress-related symptoms but with normal-borderline cut-offs at the questionnaires used in the study; Panel B shows the prevalence of children with self-perceived stress-related symptoms and with clinical cut-offs at the questionnaires used in the study.

For all children aged 1.5–5 years, scores were below the clinical cut-off for total, internalizing, and externalizing problems subscales (Fig 1 and Table S5) overall and stratified according to sociodemographic and COVID-19-related factors (Table S6 and Table S7). Among children aged 6–18 years, 7 (16.3%), 16 (37.2%), and 2 (4.6%) CBCL questionnaires exceeded the clinical cut-off for total, internalizing, and externalizing problems, respectively. Interestingly, the CBCL clinical cut-off for internalizing problems was associated with physical (p=.04), behavioural (p=.01), and cognitive (p=.03) self-perceived stress-related symptoms in children aged 6–18 years. No association between CBCL clinical cut-off for externalizing problems and self-perceived stress-related symptoms was observed in children aged 6–18 years (Fig 1 and

**Table 2. Self-perceived stress-related symptoms among children (N=77) and parents (N=81) enrolled in the study.**

| | Overall, N (%) | | | Physical, N (%) | | | | Behavioral, N (%) | | | | Emotional, N (%) | | | | Cognitive, N (%) | | | |
|---|---|---|---|---|---|---|---|---|---|---|---|---|---|---|---|---|---|---|---|
| | No | Yes | P-value | 0 | 1-2 | ≥ 3 | P-value | 0 | 1 | ≥ 2 | P-value | 0 | 1-2 | ≥ 3 | P-value | 0 | 1-2 | ≥ 3 | P-value |
| Overall (N=158) | 22 (13.9) | 136 (86.1) | – | 71 (44.9) | 42 (26.6) | 45 (28.5) | – | 110 (69.6) | 34 (21.5) | 14 (8.9) | – | 46 (29.1) | 45 (28.5) | 67 (42.4) | – | 85 (53.8) | 43 (27.2) | 30 (19) | – |
| **Familiar role** | | | | | | | | | | | | | | | | | | | |
| Children (N=77) | 11 (14.3) | 66 (85.7) | .90 | 34 (44.1) | 20 (26) | 23 (29.9) | .93 | 53 (68.8) | 20 (26) | 4 (5.2) | .16 | 21 (27.3) | 24 (31.2) | 32 (41.5) | .75 | 42 (54.5) | 22 (28.6) | 13 (16.9) | .79 |
| Parents (N=81) | 11 (13.6) | 70 (86.4) | | 37 (45.6) | 22 (27.2) | 22 (27.2) | | 57 (70.4) | 14 (17.3) | 10 (12.3) | | 25 (30.9) | 21 (25.9) | 35 (43.2) | | 43 (53.1) | 21 (25.9) | 17 (21) | |
| **Sex** | | | | | | | | | | | | | | | | | | | |
| Female (N=77) | 9 (11.7) | 68 (88.3) | .43 | 29 (37.6) | 25 (32.5) | 23 (29.9) | .15 | 54 (70.1) | 17 (22.1) | 6 (7.8) | .90 | 22 (28.5) | 21 (27.3) | 34 (44.2) | .90 | 39 (50.6) | 25 (32.5) | 13 (16.9) | .34 |
| Male (N=81) | 13 (16.1) | 68 (83.9) | | 42 (51.9) | 17 (21) | 22 (27.2) | | 56 (69.1) | 17 (21) | 8 (9.9) | | 24 (29.6) | 24 (29.6) | 33 (40.8) | | 46 (56.8) | 18 (22.2) | 17 (21) | |
| **Non-psychiatric underlying conditions** | | | | | | | | | | | | | | | | | | | |
| No (N=108) | 20 (18.5) | 88 (81.5) | **.01** | 58 (53.7) | 27 (25) | 23 (21.3) | **<.01** | 84 (77.8) | 17 (15.7) | 7 (6.5) | **<.01** | 39 (36.1) | 32 (29.6) | 37 (34.3) | **<.01** | 62 (57.4) | 30 (27.8) | 16 (14.8) | .14 |
| Yes (N=50) | 2 (4) | 48 (96) | | 13 (26) | 15 (30) | 22 (44) | | 26 (52) | 17 (34) | 7 (14) | | 7 (14) | 13 (26) | 30 (60) | | 23 (46) | 13 (26) | 14 (28) | |
| **Familiarity for neuropsychiatric conditions** | | | | | | | | | | | | | | | | | | | |
| No (N=100) | 16 (16) | 84 (84) | .32 | 51 (51) | 26 (26) | 23 (23) | .10 | 73 (73) | 21 (21) | 6 (6) | .20 | 34 (34) | 32 (32) | 34 (34) | **.01** | 58 (58) | 28 (28) | 14 (14) | .09 |
| Yes (N=57) | 6 (10.3) | 52 (89.7) | | 20 (35.1) | 16 (28.1) | 21 (36.8) | | 36 (63.2) | 13 (22.8) | 8 (14) | | 11 (19.3) | 13 (22.8) | 33 (57.9) | | 26 (45.6) | 15 (26.3) | 16 (28.1) | |
| **SES[a]** | | | | | | | | | | | | | | | | | | | |
| medium-low (N=73) | 8 (11) | 65 (89) | .26 | 30 (41.1) | 23 (31.5) | 20 (27.4) | .40 | 52 (71.2) | 19 (26) | 2 (2.8) | .06 | 23 (31.5) | 21 (28.8) | 29 (39.7) | .89 | 35 (48) | 27 (37) | 11 (15) | .52 |
| medium-high (N=81) | 14 (17.3) | 67 (82.7) | | 40 (49.4) | 18 (22.2) | 23 (28.4) | | 56 (69.1) | 15 (18.5) | 10 (12.4) | | 23 (28.4) | 23 (28.4) | 35 (43.2) | | 49 (60.4) | 16 (19.8) | 16 (19.8) | |
| **COVID-19** | | | | | | | | | | | | | | | | | | | |
| non-COVID-19 cases (N=33) | 7 (20.6) | 27 (79.4) | .10 | 22 (66.7) | 8 (24.2) | 3 (9.1) | **.01** | 28 (84.9) | 3 (9.1) | 2 (6) | .09 | 12 (36.4) | 12 (36.4) | 9 (27.2) | .14 | 19 (57.6) | 10 (30.3) | 4 (12.1) | .52 |
| COVID-19 cases (N=125) | 15 (12.1) | 109 (87.9) | | 49 (39.2) | 34 (27.2) | 42 (33.6) | | 82 (65.6) | 31 (24.8) | 12 (9.6) | | 34 (27.2) | 33 (26.4) | 58 (46.4) | | 66 (52.8) | 33 (26.4) | 26 (20.8) | |
| **Pandemic wave** | | | | | | | | | | | | | | | | | | | |

*(Continued)*

**Table 2.** (Continued)

| | Overall, N (%) | | | Physical, N (%) | | | | Behavioral, N (%) | | | | Emotional, N (%) | | | | Cognitive, N (%) | | | |
|---|---|---|---|---|---|---|---|---|---|---|---|---|---|---|---|---|---|---|---|
| | No | Yes | P-value | 0 | 1-2 | ≥ 3 | P-value | 0 | 1 | ≥ 2 | P-value | 0 | 1-2 | ≥ 3 | P-value | 0 | 1-2 | ≥ 3 | P-value |
| Parental (N=39) | 11 (27.5) | 29 (72.5) | .01 | 34 (87.2) | 4 (10.2) | 1 (2.6) | <.01 | 36 (92.3) | 3 (7.7) | 0 (0) | .01 | 14 (35.9) | 16 (41) | 9 (23.1) | <.01 | 26 (66.7) | 12 (30.8) | 1 (2.5) | .04 |
| Delta (N=59) | 5 (7.8) | 59 (92.2) | | 22 (37.9) | 16 (27.6) | 20 (34.5) | | 37 (63.8) | 14 (24.1) | 7 (12.1) | | 9 (15.5) | 20 (34.5) | 29 (50) | | 30 (51.7) | 13 (22.4) | 15 (25.9) | |
| Omicron (N=61) | 6 (11.1) | 48 (88.9) | | 15 (24.6) | 22 (36.1) | 24 (39.3) | | 37 (60.6) | 17 (27.9) | 7 (11.5) | | 23 (37.7) | 9 (14.8) | 29 (47.5) | | 29 (47.5) | 18 (29.5) | 14 (23) | |
| COVID-19 symptom | | | | | | | | | | | | | | | | | | | |
| Asymptomatic (N=12) | 10 (21.7) | 36 (78.3) | .07 | 9 (75) | 1 (8.3) | 2 (16.7) | <.01 | 9 (75) | 2 (16.7) | 1 (8.3) | .09 | 4 (33.3) | 6 (50) | 2 (16.7) | <.01 | 9 (75) | 3 (25) | 0 (0) | .01 |
| Symptomatic (N=114) | 12 (10.7) | 100 (89.3) | | 41 (36) | 33 (29) | 40 (35) | | 74 (64.9) | 29 (25.4) | 11 (9.7) | | 30 (26.3) | 27 (23.7) | 57 (50) | | 58 (50.9) | 30 (26.3) | 26 (22.8) | |
| Duration of home isolation | | | | | | | | | | | | | | | | | | | |
| <3 (N=62) | 12 (19.4) | 50 (80.6) | .11 | 22 (35.5) | 18 (29) | 22 (35.5) | .13 | 40 (64.5) | 19 (30.7) | 3 (4.8) | .05 | 27 (43.6) | 11 (17.7) | 24 (38.7) | <.01 | 30 (48.4) | 24 (38.7) | 8 (12.9) | .02 |
| ≥ 3 (N=96) | 10 (10.4) | 86 (89.6) | | 49 (51) | 24 (25) | 23 (24) | | 70 (72.9) | 15 (15.6) | 11 (11.5) | | 19 (19.8) | 34 (35.4) | 43 (44.8) | | 55 (57.3) | 19 (19.8) | 22 (22.9) | |

[a]SES = Socio-economic status;

[b]The presence of self-perceived stress-related symptoms were assessed among only COVID-19 cases.

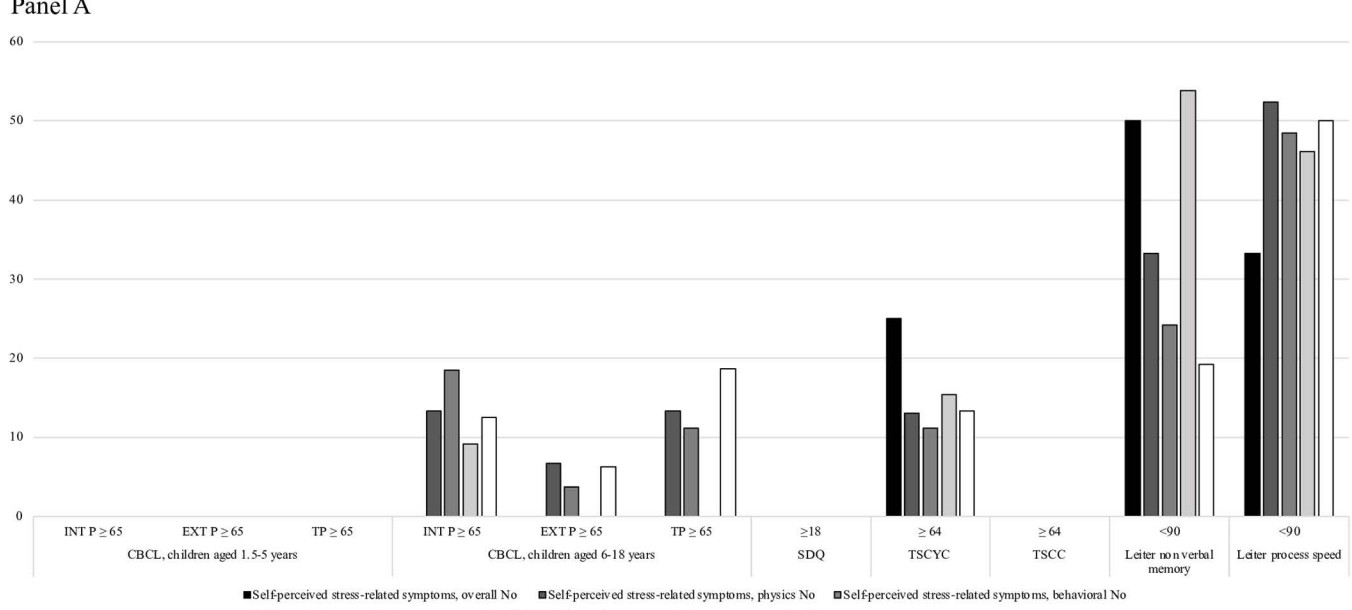

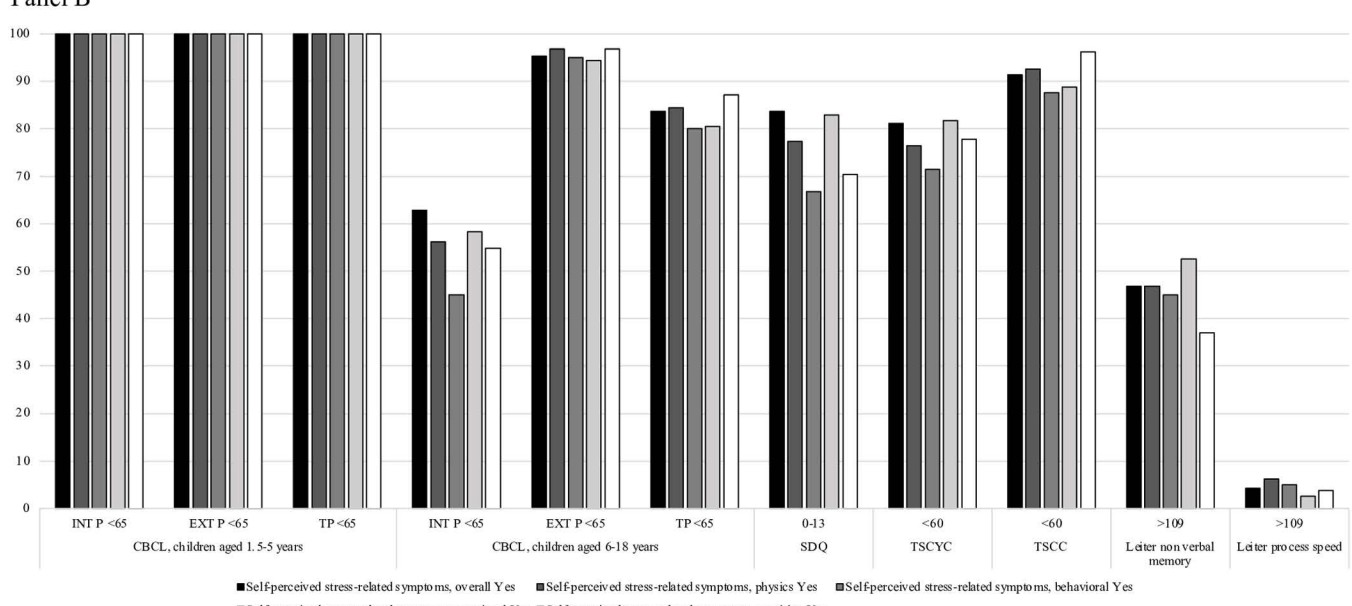

**Fig 1. Assessment of emotional-behavioral and PTSD-related symptoms in children through the CBCL, SDQ-4-17, TSCYC, and TSCC questionnaires and the Leiter-3 test. Panel A shows the relationship between the absence of self-perceived stress-related symptoms and clinical significance scores in study questionnaires. Panel B shows the relationship between the presence of self-perceived stress-related symptoms and normal-borderline scores in study questionnaires.**

Table S5). No differences were observed when patients were stratified according to sociodemographic and COVID-19-related factors (Table S6 and Table S7).

A total of 53/61 (86.9%) SDQ-4–17 questionnaires were completed for children aged 3–12 years. Out of 49 individuals with self-perceived stress-related symptoms, 83.7% (N=41), 12.2% (N=6), and 4.1% (N=2) were classified as with normal, sub-clinical, and clinical cut-offs,

respectively (Fig 1 and Table S5). Interestingly, SDQ-4–17 clinical scores correlated with the presence of self-perceived physical (p=.05), behavioural (p<.01), and cognitive (p<.01) stress-related symptoms (Fig 1 and Table S5). No associations were observed between SDQ-4–17 scores and sociodemographic and COVID-19-related factors, except for the pandemic waves, with children who experienced their COVID-19 family outbreak during the Omicron wave reporting higher SDQ-4–17 clinical scores compared to those in the Parental and Delta waves (p=.01) (Table S6 and Table S7).

A total of 57/61 (93.1%) TSCYC questionnaires were filled in by parents of children aged 3–12 years. Out of 53 individuals who showed self-perceived stress-related symptoms, 81.1% (N=43), 3.8% (N=2), and 15.1% (N=8) recorded scores within the normal, sub-clinical, and clinical cut-offs at the TSCYC questionnaire, respectively (Fig 1 and Table S5). No association was observed between TSCYC clinical scores and the presence of self-perceived stress-related symptoms (Fig 1 and Table S5). Clinical cut-offs were observed more frequently in children with at least one non-neuropsychiatric underlying condition (p=.03) and familiarity with neuropsychiatric disorders (p=.03). Similar to self-perceived stress-related and total difficulty symptoms, children who experienced COVID-19 family cluster during the Omicron wave had a higher incidence of PTSD-related symptoms compared to children who shared their COVID-19 family cluster during the Parental and Delta waves (p=.02) (Table S6 and Table S7). Conversely, no differences in the PTSD-related symptoms were observed in COVID-19 cases compared to non-COVID-19 patients; similarly, no differences were observed when data were stratified according to other sociodemographic and COVID-19-related factors (Table S6 and Table S7).

A total of 39/44 (88.6%) TSCC questionnaires were completed by children aged 8–18 years. Most of children (N=32, 91.4%) who reported self-perceived stress-related symptoms recorded normal cut-off at the TSCC questionnaire (Fig 1 and Table S5). Similarly, no differences were observed when participants were stratified according to sociodemographic and COVID-19-related factors (Table S6 and Table S7).

Out of 71 children aged 3–18 years, 53 (74.7%) underwent the Leiter-3 assessment. Out of 47 children who showed self-perceived stress-related symptoms, 46.8% (N=22) recorded "above average" score at the "Non-Verbal Memory" scale. Conversely, the "Processing Speed" score was in "average," for most of children (51.1%, N=24) (Fig 1 and Table S5). The 'Non-Verbal Memory' (p<.01) and 'Processing Speed' (p=.01) scores were more commonly reported as "above average" in children who experienced their COVID-19 family cluster during the Parental and Delta waves compared to the Omicron wave. Moreover, the 'Non-Verbal Memory' score was more commonly reported as "above average" in non-COVID-19 cases compared to COVID-19 cases (p=.04) (Table S6 and Table S7).

### 3.4 Neuropsychiatric and neuropsychological assessment of parents

We evaluated the presence of symptoms related to depression, anxiety, and stress in 73/81 (90.1%) parents (F=21) using the DASS-21 questionnaire. Overall, the depression, anxiety, and stress scores of the DASS-21 exceeded clinical cut-offs in 9 (12.3%), 8 (11%), and 16 (21.9%) parents, respectively (Fig 2 and Table S8). Clinical DASS-21 depression scores were correlated with the presence of self-perceived physical (p=.02) stress-related symptoms in adults (Table S8).

Stress subscale scores were higher in parents with familiarity with neuropsychiatric conditions (p=.02) (Table S9). No differences in DASS-21 scores were observed between COVID-19 cases and non-COVID-19 cases; however, among COVID-19 cases, clinical cut-offs for the anxiety DASS-21 were more common in cases with symptomatic COVID-19 compared to those with asymptomatic COVID-19 (p=.04) (Table S9).

## Panel A

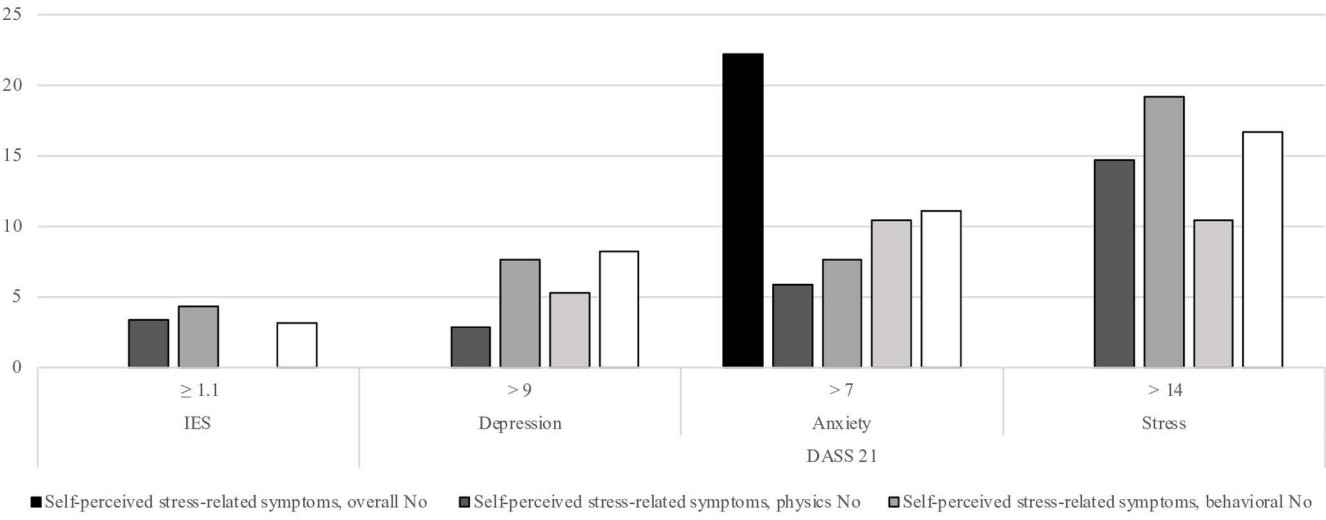

■ Self-perceived stress-related symptoms, overall No    ■ Self-perceived stress-related symptoms, physics No    ■ Self-perceived stress-related symptoms, behavioral No

□ Self-perceived stress-related symptoms, emotional No    □ Self-perceived stress-related symptoms, cognitive No

## Panel B

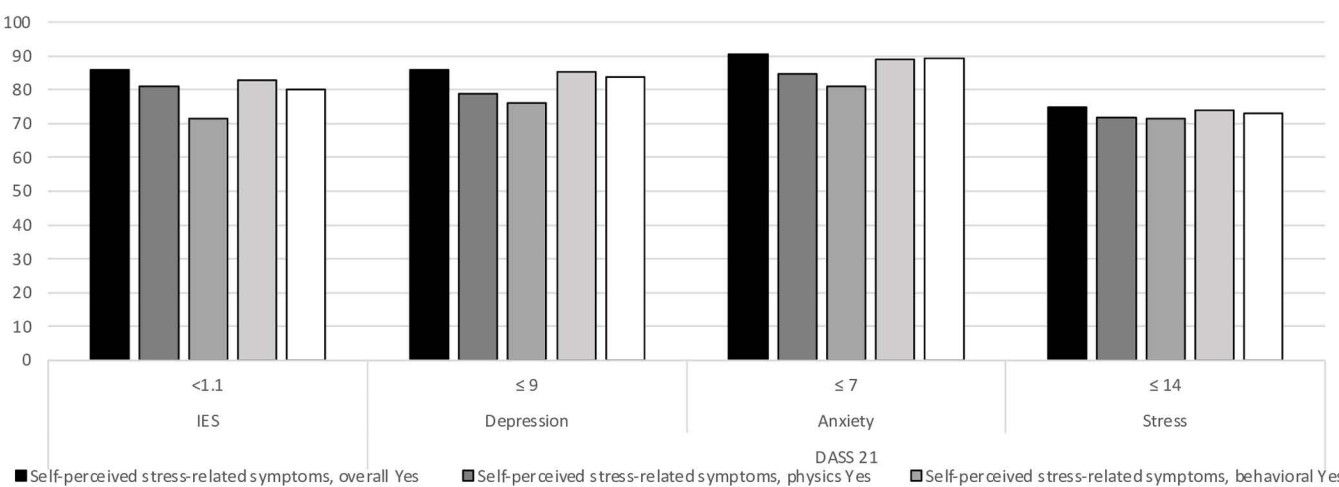

■ Self-perceived stress-related symptoms, overall Yes    ■ Self-perceived stress-related symptoms, physics Yes    ■ Self-perceived stress-related symptoms, behavioral Yes

□ Self-perceived stress-related symptoms, emotional Yes    □ Self-perceived stress-related symptoms, cognitive Yes

**Fig 2. Neuropsychiatric and neuropsychological assessment of parents. Panel A shows the relationship between the absence of self-perceived stress-related symptoms and clinical significance scores in study questionnaires. Panel B shows the relationship between the presence of self-perceived stress-related symptoms and normal-borderline scores in study questionnaires.**

Moreover, we evaluated the self-perceived presence of symptoms related to a traumatic event using the IES-R questionnaire, which was completed by 66/81 (74.1%) parents [F=37]. The majority of parents (N=58, 87.9%) scored within the normal cut-off, with only 8 (12.1%) parents receiving a clinical score, with no differences between mothers and fathers (Table S9). Interestingly, clinical IES-R scores correlated with the presence of self-perceived physical (p=.04), behavioural (p<.01), emotional (p=.05), and cognitive (p=.04) stress-related

symptoms in parents (Fig 2 and Table S9). Familiarity with neuropsychiatric underlying conditions (p=.04) and experiencing a COVID-19 family outbreak during the Omicron wave (p=.01) were associated with a clinical IES-R score. Conversely, no differences were observed between COVID-19 cases and non-COVID-19 cases (Table S9).

### 3.5 Stress-related, emotional-behavioural, and PTSD-related symptoms in COVID-19 family clusters

Among 46 enrolled families, more than half (N=39, 92.9%) had at least one child and one parent with self-perceived stress-related symptoms.

Fig 3 reveals a positive correlation between the number of self-perceived stress-related symptoms in children and their parents within the same family (r=.53, p=.0005).

Regarding the standardized assessment of stress-related symptoms, IES-R scores of mothers correlated with the mean TSCYC scores of their young children (r=.47, p=.01) but did not correlate with the mean TSCC scores of older children. Conversely, IES-R scores of fathers were not correlated with the mean TSCYC or TSCC scores of their children (Table S10).

In the emotional-behavioural assessment of children and parents, total problems for children aged 1.5–5 years correlated with stress DASS-21 scores in parents (r=.56, p=.02). Similarly, the internalizing problems subscale for children aged 1.5–5 years correlated with

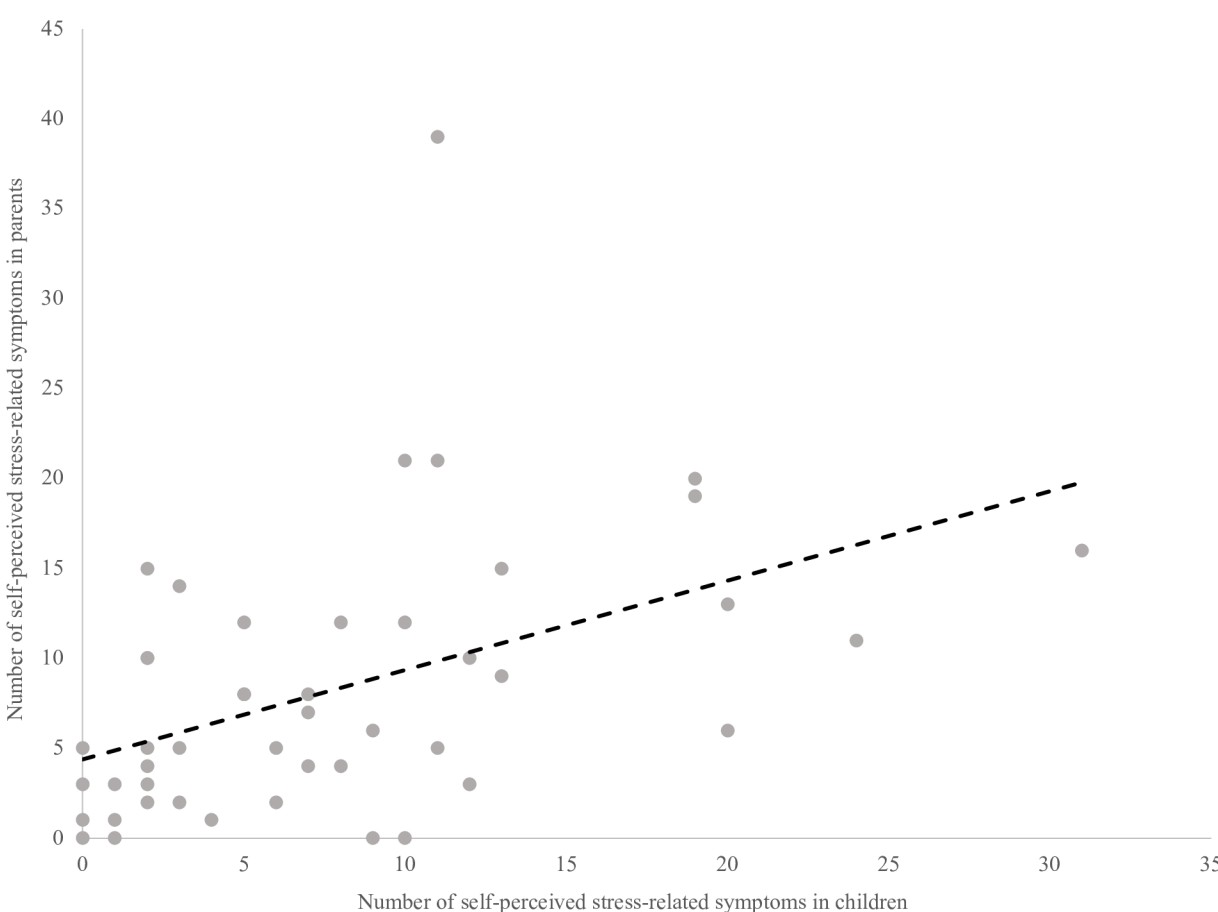

**Fig 3. Correlation of self-perceived stress-related symptoms among children and their parents within the same COVID-19 family clusters.**

anxiety DASS-21 scores in parents (r=.56, p=.02). Conversely, externalizing problems did not correlate with any DASS-21 score. Regarding children aged 6–18 years, their CBCL total problems scores correlated with anxiety DASS-21 scores in parents (r=.39, p=.02). Internalizing problems correlated positively with depression (r=.38, p=.03), anxiety (r=.49, p<.01), and stress (r=.42, p=.01) DASS-21 scores in parents (Table S11).

## 4. Discussion

We investigated the prevalence and characteristics of neuropsychiatric and neuropsychological sequelae following COVID-19 in a cohort of Italian COVID-19 family clusters.

Our findings revealed the presence of self-perceived emotional-behavioural and stress-related symptoms in both children and adults following COVID-19. In particular, our results documented a positive correlation of stress-related symptoms among children and their parents within the same family, suggesting potential mutual influence on increasing stress-related symptoms between generations.

The unique nature of our cohort of COVID-19 family clusters enabled us to comprehensively evaluate the neuropsychiatric and neuropsychologic impact of the COVID-19 pandemic, exploring differences between children and adults. Furthermore, analyzing family clusters with COVID-19 and non-COVID-19 cases allowed us to explore the neuropsychiatric impact of COVID-19 infection and social restrictions adopted during the pandemic among infected and uninfected participants. Additionally, as more than 95% of our cohort consisted of individuals with mildly symptomatic COVID-19, it closely mirrors the European epidemiological situation, where only 0.9% of the overall cases developed a severe illness.[35] Furthermore, our study population included children with no prior history of neuropsychiatric disorders, making it representative of mental healthy individuals facing the challenges posed by the pandemic.

Our assessment of self-perceived stress-related symptoms revealed that nearly 90% of our participants reported at least one symptom, with approximately one-third experiencing more than three, indicating a high-stress level among participants. Emotional and physical symptoms were the most commonly reported, affecting over 70% and 50% of our population, respectively. Notably, the presence of at least one non-neuropsychiatric underlying condition was associated with higher physical stress-related symptoms, while a familial history of neuropsychiatric conditions was linked to increased emotional stress symptoms. These findings align with previous data demonstrating a higher prevalence of neuropsychiatric impact of the COVID-19 pandemic in children with underlying conditions other than neuropsychiatric diagnoses.[36]

Interestingly, the presence of self-perceived stress-related symptoms contrasted with a prior study conducted at our institution, which employed the same self-report questionnaire for stress-related symptoms and reported lower frequencies of stress-related symptoms in patients with psychiatric disorders shortly after the pandemic.[26]

Differences in the frequency of self-perceived stress-related symptoms were not observed between parents and children, suggesting that participants were similarly affected by the pandemic, regardless of age. Conversely, a higher prevalence of self-perceived stress-related symptoms was observed in COVID-19 cases compared to non-COVID-19 cases. These findings are in line with previous studies suggested a higher risk for mental health sequelae in infected compared to uninfected subjects.[5,10]

When examining PTSD-related symptoms, most children displayed no post-traumatic stress symptoms, regardless of age. Notably, our findings differ from a prior study that found a high percentage of PTSD-related symptoms, particularly in children aged 7–10 years.[37]

We assessed emotional-behavioural changes in children, documenting that internalizing problems were the most frequent issues in youths aged 6–18 years, with 36% of participants affected, followed by externalizing problems in nearly 25% of individuals. These findings were consistent with existing literature, which shows a higher prevalence of internalizing problems among children and adolescents post-COVID-19.[38,39]

Conversely, few children showed clinical neuropsychological alterations on the Leiter-3 test. These findings were in line with a previous study conducted on COVID-19 family clusters, showing a prevalence of neuropsychological sequelae of 8.1%.[40] However, they contrasted with previous findings indicating a decline in neuropsychological performance post-COVID-19 in children.[39,41]

When stratifying emotional-behavioral and PTSD-related symptoms according to COVID-19-related factors, no differences were observed between COVID-19 and non-COVID-19 cases in both children and parents. This was in contrast to what we observed for self-perceived stress-related symptoms and may suggest a possible key role of social restrictions in the onset of neuropsychiatric and neuropsychological problems, in parallel to the impact of SARS-CoV-2 infection, as already hypothesized by other authors. [42,43]

Notably, children and parents who experienced their COVID-19 family outbreak during the Omicron wave showed a higher prevalence of emotional-behavioural and stress-related symptoms, suggesting that a shorter timeframe between infection and/or isolation and the neuropsychiatric assessment may lead to more pronounced neuropsychiatric symptoms in our cohort. Similarly, other findings showed a decrease in neuropsychological impact of COVID-19 over time.[44]

Lastly, we documented a positive correlation in the number of self-perceived stress-related symptoms between children and their parents within the same family, suggesting that an increase in stress symptoms in parents following COVID-19 infection and/or isolation may be associated with an increase in stress symptoms in children, and vice versa. This finding is consistent with existing literature, highlighting the mutual influence between children's and parents' stress levels.[45,46] Parents' individual and dyadic stress may mediate the impact of isolation on children's behavioral and emotional problems.[47] Furthermore, PTSD-related symptoms in mothers correlated with PTSD-related symptoms in their younger children. This correlation could be attributed to the predominant role of mothers in serving as the primary caregivers, particularly for younger children.[46]

The prospective and systematic direct examination of families is a strength of this study, expanding on previous studies primarily based on online surveys and pediatric cohorts. However, our study has several limitations. Firstly, the limited number of enrolled participants may lead to underestimating neuropsychiatric and neuropsychological symptoms, especially in children. Additionally, the absence of a control group of families that did not experience a COVID-19 household outbreak prevented us from evaluating the impact of isolation on children and their parents. Furthermore, the lack of a control group of hospitalized patients for COVID-19 hindered us from assessing the neuropsychiatric impact of severe infection and hospitalization. Moreover, we lacked pre-pandemic neurocognitive and neuropsychological assessments of participants, possibly resulting in the misclassification of symptoms. Finally, with regard to the questionnaires that were meant to be completed by parents, some were filled out by mothers while others were completed by fathers, which may have introduced variability. Previous research has shown that fathers and mothers may differ in their reporting of certain behaviors, particularly for externalizing problems.[48]

In conclusion, our study offers a comprehensive assessment of the neuropsychiatric and neuropsychological impact of the COVID-19 pandemic on families. It underscores the pandemic's effect on stress in both children and adults, as well as the reciprocal influence of stress symptoms

within families. Further research, including extensive neuropsychological and neuropsychiatric assessments within family clusters, is needed to gain a deeper understanding of the long-term consequences for children and their parents, paving the way for support strategies to mitigate the potential adverse effects of the COVID-19 pandemic. A more comprehensive evaluation should involve the full Leiter-3 battery to assess children's cognitive performance, alongside the Youth Self Report (YSR) and the CBCL for a nuanced comparison of emotional and behavioral issues, with self-report questionnaires completed by both parents when possible.

## Supporting information

**Table S1. Neuropsychiatric assessment for children and their parents at the COVID-19 family cluster clinic (CovFC).**
(DOCX)

**Supplementary Questionnaire. Self-administered questionnaire assessing stress-related symptoms in children and parents.**
(DOCX)

**Table S2. Clinical cut-off for the Strengths and Difficulties Questionnaire (SDQ 4–17) subscales.**
(DOCX)

**Table S3. Clinical cut-off for the Trauma Symptom Checklist for Young Children (TSCYC) and the Trauma Symptom Checklist for Children (TSCC) subscales.**
(DOCX)

**Table S4. Self-perceived stress-related symptoms among children (N=77) and parents (N=81) enrolled in the study.**
(DOCX)

**Table S5. Emotional-behavioral and PTSD-related symptoms in children according to socio-demographic factors, assessed with the CBCL, SDQ-4–17, TSCYC, and TSCC questionnaires.**
(DOCX)

**Table S6. Emotional-behavioral and PTSD-related symptoms in children according to COVID-19-related factors, assessed with the CBCL, SDQ-4–17, TSCYC, and TSCC questionnaires.**
(DOCX)

**Table S7. Neuropsychiatric and neuropsychological symptoms in enrolled parents according to sociodemographic and COVID-19-related factors.**
(DOCX)

**Table S8. Correlation of the self-perceived stress-related symptoms between children and their parents belonging the same COVID-19 family cluster.**
(DOCX)

**Table S9. Correlation between stress-related symptoms in children and their parents belonging the same COVID-19 family cluster, r (p-value).**
(DOCX)

**Table S10. Correlation between stress-related symptoms in children and their parents belonging the same COVID-19 family cluster, r (p-value).**
(DOCX)

**Table S11. Correlation between emotional-behavioral symptoms in children and their parents belonging the same COVID-19 family cluster, r (p-value).**
(DOCX)

## Acknowledgements

The corresponding author would like to thank Dr. Sandra Cozzani and Dr. Bertilla Ranzato for their support in patients' enrollment. The authors thank all the family paediatricians collaborating with the project. The authors thank all families who attended the CovFC of the University Hospital of Padova.

## Author contributions

**Conceptualization:** Costanza Di Chiara, Martina Ferrarese, Alessia Raffagnato, Carlo Giaquinto, Michela Gatta.

**Data curation:** Costanza Di Chiara, Martina Ferrarese, Riccardo Boracchini, Anna Cantarutti, Federica Occhino.

**Formal analysis:** Riccardo Boracchini, Anna Cantarutti.

**Funding acquisition:** Carlo Giaquinto.

**Investigation:** Costanza Di Chiara, Martina Ferrarese, Anna Letizia Tibaldo, Chiara Stefanni, Alessia Raffagnato, Federica Occhino.

**Methodology:** Costanza Di Chiara, Martina Ferrarese, Alessia Raffagnato, Carlo Giaquinto, Michela Gatta.

**Project administration:** Costanza Di Chiara.

**Resources:** Costanza Di Chiara, Marica De Pieri.

**Supervision:** Carlo Giaquinto, Michela Gatta.

**Validation:** Daniele Donà, Benedetta Tascini, Marina Miscioscia, Michela Gatta.

**Visualization:** Daniele Donà, Benedetta Tascini, Marina Miscioscia.

**Writing – original draft:** Costanza Di Chiara, Martina Ferrarese.

**Writing – review & editing:** Riccardo Boracchini, Anna Cantarutti, Anna Letizia Tibaldo, Chiara Stefanni, Daniele Donà, Marica De Pieri, Alessia Raffagnato, Benedetta Tascini, Marina Miscioscia, Federica Occhino, Carlo Giaquinto, Michela Gatta.

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
