## [Decision Letter · Decision Letter 0]

27 May 2024

PONE-D-23-36142Neuropsychiatric and neuropsychological impact of the pandemic in Italian COVID-19 family clusters, including children and parentsPLOS ONE

Dear Dr. Di Chiara,

Thank you for submitting your manuscript to PLOS ONE. After careful consideration, we feel that it has merit but does not fully meet PLOS ONE’s publication criteria as it currently stands. Therefore, we invite you to submit a revised version of the manuscript that addresses the points raised during the review process.

We look forward to receiving your revised manuscript.

Kind regards,

Humayun Kabir

Academic Editor

PLOS ONE

3. In the online submission form, you indicated that [The clinical data of the current manuscript are available from the corresponding author upon reasonable request.]. 

4. review your reference list to ensure that it is complete and correct. If you have cited papers that have been retracted, please include the rationale for doing so in the manuscript text, or remove these references and replace them with relevant current references. Any changes to the reference list should be mentioned in the rebuttal letter that accompanies your revised manuscript. If you need to cite a retracted article, indicate the article’s retracted status in the References list and also include a citation and full reference for the retraction notice.

Additional Editor Comments: 

Reviewers' comments:

Reviewer's Responses to Questions

**Comments to the Author**

1. Is the manuscript technically sound, and do the data support the conclusions?

Reviewer #1: Yes

2. Has the statistical analysis been performed appropriately and rigorously? 

Reviewer #1: Yes

3. Have the authors made all data underlying the findings in their manuscript fully available?

Reviewer #1: Yes

4. Is the manuscript presented in an intelligible fashion and written in standard English?

Reviewer #1: Yes

5. Review Comments to the Author

Reviewer #1: In "introduction" section, the authors cited that there is in the literature a gap regarding the prevalence and nature of neuropsychiatric and neuropsychological sequelae following COVID-19 in children and adolescents and, only few studies have reported this topic. It is noteworthy that also studies on Covid 19-impact have been conducted in a specific pediatric population, such as children and adolescents with neuropsychiatric disorders (Operto et al). The authors found a statistically significant increase of internalizing problems, anxiety, depression and sleep problems in analyzed sample, with small differences between ages, and high levels of parental stress.

These studies confirm that social isolation and restrictions related to lockdown, and the onset of a new infectious disease have had a significant impact on children’s well-being and their families.

In this context, this manuscript is placed, focusing on the prevalence and nature of neuropsychiatric and neuropsychological sequelae following Covid-19 exposure in families and expanding the research on this topic.

Results section:

-The authors cited that participants who lacked analytical evidence of SARS-CoV-2 infection were designated as "nonCOVID-19" cases. Why they were included as asymptomatic ("severity of COVID-19" section) in Table 1?

-In table 2 (and tables in supplemental materials), the statistically significant p-value should be highlighted in bold or with other mark, to make the correlation clearer

-The figures are not clear enough to display correctly due to their blurriness.

6. PLOS authors have the option to publish the peer review history of their article (what does this mean? ). If published, this will include your full peer review and any attached files.

**Do you want your identity to be public for this peer review?** For information about this choice, including consent withdrawal, please see our Privacy Policy .

Reviewer #1: **Yes: ** Francesca Felicia Operto

---

## [Author Response · Author response to Decision Letter 0]

6 Aug 2024

Padua, June 4, 2024

PONE-D-23-36142

Neuropsychiatric and neuropsychological impact of the pandemic in Italian COVID-19 family clusters, including children and parents.

Dear Editor and Reviewers of the PLOS ONE Journal,

We thank you for your interest and comments on our manuscript.

We tried to answer the queries at our best. Enclosed you find our replies.

We are pleased to resubmit our manuscript after revision, according to the Academic Editor and Reviewer’s comments.

Costanza Di Chiara, on behalf of the all authors 

Editorial comments

Reply: We have reviewed our manuscript style following the PLOS ONE author guidelines you provided.

Reply: We apologize for the error. We confirm that no funding have been received for this manuscript.

3. In the online submission form, you indicated that [The clinical data of the current manuscript are available from the corresponding author upon reasonable request.].

Reply: We confirm that clinical data associated with our study are accessible solely upon reasonable request from the corresponding author. This is in line with the ethical considerations and requirements set forth by our institutional review board and with the informed consent that has been signed by participants/caregivers. The data were collected and stored in a pseudo-anonymized format at the Department for Child and Women's Health of the University of Padova, which is the corresponding author's affiliation. We employed the REDCAP© platform, developed by Vanderbilt University, Tennessee, for this purpose. Upon receiving a motivated request, the corresponding author will provide interested parties with anonymized data. Indeed, to enhance patient privacy, dates of birth will be converted into age ranges before the data are shared.

4. review your reference list to ensure that it is complete and correct. If you have cited papers that have been retracted, please include the rationale for doing so in the manuscript text, or remove these references and replace them with relevant current references. Any changes to the reference list should be mentioned in the rebuttal letter that accompanies your revised manuscript. If you need to cite a retracted article, indicate the article’s retracted status in the References list and also include a citation and full reference for the retraction notice.

Reply: The reference list has been thoroughly reviewed and updated to incorporate the latest reference, #21. Additionally, the citation style has been carefully reviewed to ensure adherence to the authors' guidelines. 

Reply to Reviewer

We have summarized here the amendments made to the manuscript based on the reviewer's comments.

Introduction:

1. In "introduction" section, the authors cited that there is in the literature a gap regarding the prevalence and nature of neuropsychiatric and neuropsychological sequelae following COVID-19 in children and adolescents and, only few studies have reported this topic. It is noteworthy that also studies on Covid 19-impact have been conducted in a specific pediatric population, such as children and adolescents with neuropsychiatric disorders (Operto et al). The authors found a statistically significant increase of internalizing problems, anxiety, depression and sleep problems in analyzed sample, with small differences between ages, and high levels of parental stress. These studies confirm that social isolation and restrictions related to lockdown, and the onset of a new infectious disease have had a significant impact on children’s well-being and their families.

In this context, this manuscript is placed, focusing on the prevalence and nature of neuropsychiatric and neuropsychological sequelae following Covid-19 exposure in families and expanding the research on this topic.

Reply: We thank the Reviewer for their comment. As suggested, we have included a sentence in the main text highlighting the existence of evidence on the impact of COVID-19 and social restrictions on the well-being of parents and the correlation of parental and familiar stress with the emotional/behavioural worsening of children and adolescents with underlying neuropsychiatric disorders. We have incorporated reference #21 [Operto FF, Scaffidi Abbate C, Piscitelli FT, et al. Adolescents with Neuropsychiatric Disorders during the COVID-19 Pandemic: Focus on Emotional Well-Being and Parental Stress. Healthcare (Basel). 2022;10(12):2368. Published 2022 Nov 25. doi:10.3390/healthcare10122368] into the main text accordingly.

Changes: lines 111-114: The social restrictions imposed during the pandemic have notably disrupted daily family routines, increasing stress in children and parents. This stress potentially leads to the deterioration of children's mental health as well as exacerbates emotional and behavioral symptoms in adolescents with underlying neuropsychiatric disorders.21

Results section:

1. The authors cited that participants who lacked analytical evidence of SARS-CoV-2 infection were designated as "nonCOVID-19" cases. Why they were included as asymptomatic ("severity of COVID-19" section) in Table 1?

Reply: We appreciate the reviewer's feedback. Initially, in Table 1, we included non-COVID-19 cases labeled as asymptomatic, confirmed by negative SARS-CoV-2 IgG test results at enrollment, to highlight their lack of symptoms during their COVID-19 family cluster. However, upon reflection, we acknowledge that this detail is redundant considering its clear explanation in the text. Consequently, we have removed the count of asymptomatic non-COVID cases from Table 1.

Changes: Please refer to Table 1 in the main text. The counts of asymptomatic non-COVID-19 cases have been removed from the 'overall', 'children', and 'parents' columns.

2. In table 2 (and tables in supplemental materials), the statistically significant p-value should be highlighted in bold or with other mark, to make the correlation clearer

Reply: We appreciate the reviewer's suggestion. Statistically significant p-values have been highlighted in bold in all main and supplemental tables.

Changes: please refer to the tables in both the main text and the supplemental material documents.

3. The figures are not clear enough to display correctly due to their blurriness.

Reply: We appreciate the Reviewer for bringing this matter to our attention, and we apologize for the poor quality of the uploaded graph. We have saved them using a different method, and they should now be legible.

Change: Please refer to the uploaded graphs for the necessary changes.

---

## [Decision Letter · Decision Letter 1]

9 Dec 2024

PONE-D-23-36142R1Neuropsychiatric and neuropsychological impact of the pandemic in Italian COVID-19 family clusters, including children and parentsPLOS ONE

Dear Dr. Di Chiara,

Thank you for submitting your manuscript to PLOS ONE. After careful consideration, we feel that it has merit but does not fully meet PLOS ONE’s publication criteria as it currently stands. Therefore, we invite you to submit a revised version of the manuscript that addresses the points raised during the review process.

We look forward to receiving your revised manuscript.

Kind regards,

Bao-Liang Zhong

Academic Editor

PLOS ONE

Journal Requirements:

Reviewers' comments:

Reviewer's Responses to Questions

**Comments to the Author**

1. If the authors have adequately addressed your comments raised in a previous round of review and you feel that this manuscript is now acceptable for publication, you may indicate that here to bypass the “Comments to the Author” section, enter your conflict of interest statement in the “Confidential to Editor” section, and submit your "Accept" recommendation.

Reviewer #1: All comments have been addressed

Reviewer #2: (No Response)

Reviewer #3: All comments have been addressed

2. Is the manuscript technically sound, and do the data support the conclusions?

Reviewer #1: Yes

Reviewer #2: No

Reviewer #3: Yes

3. Has the statistical analysis been performed appropriately and rigorously? 

Reviewer #1: Yes

Reviewer #2: No

Reviewer #3: Yes

4. Have the authors made all data underlying the findings in their manuscript fully available?

Reviewer #1: Yes

Reviewer #2: Yes

Reviewer #3: Yes

5. Is the manuscript presented in an intelligible fashion and written in standard English?

Reviewer #1: Yes

Reviewer #2: Yes

Reviewer #3: Yes

6. Review Comments to the Author

Reviewer #1: (No Response)

Reviewer #2: This study investigated the long-term effetct of pandemic COVD-19 on neuropsychiatric and neuropsychological impact in 46 Italian families, including children and parents, attending the COVID-19 Follow-up Clinic at the Department for Women’s and Children’s Health,

Padua (Italy) from December 2021 to November 2022.

A total of 81 parents and 77 children were asessed for Self-perceived stress-related,

emotional-behavioral, and post-traumatic stress (PTSD-related) symptoms. Overall,125 (79%) and 33 (21%) participants were classified as COVID-19 cases and non-COVID-19 cases, respectively.

The time interval between the COVID-19 family outbreak and the neuropsychiatric and psychological assessment

was ≤4 months for 89 (56.3%) participants and >4 months for 69 (43.8%) participants.

Authors conclude that while a higher prevalence of self-perceived stress-related symptoms

was found in COVID-19 cases compared to non-COVID-19 cases (p=.01), no

differences were observed for emotional-behavioral and PTSD-related symptoms

between the two groups.

The study is interesting and may stir interest in the reader. Nonethelss there are major criticisms about this study authors should deal with:

- Abstract: it is not enough clear if this data actually assess the long term effects ; why long-term if the data were gathered less than four months before COVI in many families; and how long the follow up was after 4 months?

- The term long term should be stressed also in the title

- It is not clear which were the mild or moderate/severe symptoms these patients developed. These should be clarified in results and discussion

- It is clear that this study is too much limited by a small intrafamilies no covid group and by the absence of a control group of families with no-COVID and isolation only in the same geographic area.

- It is not as well clear in the text the effect of the interval from COVID onset and neuropsichological/psychiatric assesment: was a longer interval (> 4 months) associated with a stress reduction?

- Authors should try to comment on higher non verbal memory functioning and precessing speed found in 5% of children from COVID family clusters due to Omicron vs Delta virus.

Reviewer #3: The author has address all the comments from the previous reviewer. However, I still have further comments as below:

2.4. Statistical analysis, line 242, statement about the software should be in past tense.

In the study limitations, line 503, you mentioned about limited number of participants, have not you calculated the sample size and study power prior to the study?

Also in line 512, please explain what would be the impact of either mother or father who completed the questionnaire.

In recommendation about further research, line 517, what is entail in the extensive neuropsychological and neuropsychiatric assessments within family clusters to be included in further study?

Table 1, the row that has all zero value (Clinical classification: critical and MIS-C, post acute sequelae) can be omitted and put as narrative, also Figure 1, panel A. The graphic on CBCL for children aged 1.5-5 years contained no value, it could be omitted and just put as narrative.

7. PLOS authors have the option to publish the peer review history of their article (what does this mean? ). If published, this will include your full peer review and any attached files.

**Do you want your identity to be public for this peer review?** For information about this choice, including consent withdrawal, please see our Privacy Policy .

Reviewer #1: No

Reviewer #2: No

Reviewer #3: No

---

## [Author Response · Author response to Decision Letter 1]

24 Jan 2025

Padova, January 23, 2025

PONE-D-23-36142R1

Neuropsychiatric and neuropsychological impact of the pandemic in Italian COVID-19 family clusters, including children and parents.

Dear Editor and Reviewers of the PLOS ONE Journal,

We thank you for your interest and comments on our manuscript.

We tried to answer the queries at our best. Enclosed you find our replies.

We are pleased to resubmit our manuscript after revision, according to the Academic Editor and Reviewers’ comments.

Costanza Di Chiara, on behalf of the all authors 

Reply to Reviewers

We have summarized here the amendments made to the manuscript based on the reviewers' comments.

Reviewer #1: no comments provided.

Reviewer #2:

This study investigated the long-term effetct of pandemic COVD-19 on neuropsychiatric and neuropsychological impact in 46 Italian families, including children and parents, attending the COVID-19 Follow-up Clinic at the Department for Women’s and Children’s Health, Padua (Italy) from December 2021 to November 2022.

A total of 81 parents and 77 children were assessed for Self-perceived stress-related,

emotional-behavioral, and post-traumatic stress (PTSD-related) symptoms. Overall,125 (79%) and 33 (21%) participants were classified as COVID-19 cases and non-COVID-19 cases, respectively.

The time interval between the COVID-19 family outbreak and the neuropsychiatric and psychological assessment was ≤4 months for 89 (56.3%) participants and >4 months for 69 (43.8%) participants.

Authors conclude that while a higher prevalence of self-perceived stress-related symptoms

was found in COVID-19 cases compared to non-COVID-19 cases (p=.01), no differences were observed for emotional-behavioral and PTSD-related symptoms between the two groups.

The study is interesting and may stir interest in the reader. Nonethelss there are major criticisms about this study authors should deal with:

-abstract: it is not enough clear if this data actually assess the long term effects ; why long-term if the data were gathered less than four months before COVI in many families; and how long the follow up was after 4 months?

Reply: We thank the Reviewer for this comment. Among the 89 patients evaluated ≤4 months post-COVID-19, the median time between infection and the neuropsychological and neuropsychiatric assessments was 3 months (IQR = 0). This underscores that the majority were evaluated at 3 months post-COVID-19, aligning with the current definition of long COVID in both children and adults, which specifies 3 months as the time frame for assessing the persistence or emergence of new symptoms following the initial SARS-CoV-2 infection. For the 69 participants assessed >4 months post-COVID-19, the median time between infection and clinical evaluation was 11.5 months (IQR = 5–12). Therefore, we believe it is appropriate to refer to long-term evaluation (and effect) in this manuscript. To provide greater clarity on the median evaluation time for these two subgroups (i.e., ≤4 months and >4 months), we have updated the abstract, title, and manuscript as shown below.

Changes:

- Title: lines 4-5: Long-term neuropsychiatric and neuropsychological impact of the pandemic in Italian COVID-19 family clusters, including children and parents.

- abstract: lines 61-64: The time interval between the COVID-19 family outbreak and the neuropsychiatric and psychological assessment was ≤4 months (median=3 months [IQR=0]) for 89 (56.3%) participants and >4 months for 69 (43.8%) (median=11.5 months [IQR=5-12]) participants.

- manuscript: lines 268-271: Specifically, the median time between the baseline and the neuropsychological assessment was 3 (IQR=3-3) for those assessed within ≤4 months, and 11.5 (IQR=5-12) months for those assessed after >4 months

- The term long term should be stressed also in the title

Reply: For this change, please, see the answer above.

- It is not clear which were the mild or moderate/severe symptoms these patients developed. These should be clarified in results and discussion

Reply: we thank the Reviewer for their comment. We have added the frequency of symptoms among symptomatic children in our cohort as a footnote in Table 1, which shows the COVID-19-related clinical characteristics of children and parents enrolled in the study. However, symptoms of mild COVID-19 are not available for parents. Indeed, COVID-19 was defined as asymptomatic, mild, moderate, or severe by clinical investigators in the team, according to the WHO clinical classification, as mentioned in the methods section (lines 218-219). Therefore, while for children we did collect more granular details on the acute symptoms of COVID-19, for their parents, we can only report the clinical severity of COVID-19, as well as the level of required care and outcomes. As per WHO classification, moderate to severe COVID-19 is defined as the presence of radiologically confirmed pneumonia (moderate) associated with respiratory distress and/or oxygen desaturation (severe). In the results section, it’s been specified that moderate/severe COVID-19 was characterized by the presence of pneumonia requiring hospitalization.

Changes:

- Table 1 - footnote c: Among children, the most common symptoms were fever (64.9%), rhinitis (35.1%), headache (24.6%), asthenia (17.5%), and cough (15.8%).

- Lines 264-265: Among the 81 parents, 60 (74.1%) were COVID-19 cases, with 4 (6.7%) having asymptomatic and 53 (88.3%) mild COVID-19, while 3 (5%) parents experienced moderate/severe infection characterized by radiologically confirmed pneumonia requiring hospitalization.

- It is clear that this study is too much limited by a small intrafamilies no covid group and by the absence of a control group of families with no-COVID and isolation only in the same geographic area.

Reply: We agree with the Reviewer that this is one of the limitations of this study. Indeed, we have acknowledged it at the bottom of the discussion section, among other limitations. The Reviewer can find this sentence in lines 511-513 “Additionally, the absence of a control group of families that did not experience a COVID-19 household outbreak prevented us from evaluating the impact of isolation on children and their parents.”

- It is not as well clear in the text the effect of the interval from COVID onset and neuropsichological/psychiatric assesment: was a longer interval (> 4 months) associated with a stress reduction?

Reply: We would like to clarify that the classification of patients assessed at ≤4 months and >4 months was included solely to indicate the number of participants evaluated at medium- and long-term time points post-infection. This classification highlights those who, as mentioned above, met the criteria for persistent symptoms post-COVID. However, as stated in the manuscript, the primary focus of this study was to evaluate the impact of the length of isolation (< vs. >3 weeks) and the infecting SARS-CoV-2 variant of concern on the development of neuropsychological symptoms. Notably, the latter also indirectly reflects the impact of time on symptoms, as the different variants of concern correspond to the time periods during which participants developed.

- Authors should try to comment on higher non verbal memory functioning and precessing speed found in 5% of children from COVID family clusters due to Omicron vs Delta virus.

Reply: To our knowledge, no study in the current literature has investigated potential differences in neuropsychological performance between patients infected with different SARS-CoV-2 variants of concern (VOC). In our cohort, the small difference observed—specifically, higher non-verbal memory functioning and processing speed in children who experienced a COVID-19 family cluster during the Delta wave compared to the Omicron wave—could be explained by the longer recovery period (both physical and emotional) available to those affected during the Delta wave. This may have influenced the neuropsychological evaluation, as such assessments are well-known to be impacted by emotional stressors, as discussed in the discussion section. However, we acknowledge that this interpretation is limited by the small number of participants, particularly those with higher scores. The limited sample size has already been addressed in the manuscript’s limitations section. 

Reviever #3:

The author has address all the comments from the previous reviewer. However, I still have further comments as below:

- 2.4. Statistical analysis, line 242, statement about the software should be in past tense.

Reply: we thank the Reviewer for catching this. The sentence has been changed to past tense.

Changes: lines 243-244: “will be performed” has been changed to “were performed”.

- In the study limitations, line 503, you mentioned about limited number of participants, have not you calculated the sample size and study power prior to the study?

Reply: Thank you for your valuable feedback. First of all, it is important to clarify that this study was primarily descriptive in nature, aiming to observe and characterize the perceived stress-related, emotional-behavioral, and post-traumatic stress symptoms in a cohort of family clusters. While a formal sample size calculation was not performed prior to the study, we acknowledge that a larger sample size could have provided greater statistical power and enhanced the generalizability of our findings. The limited sample size has been acknowledged as a potential limitation of the study.

- Also in line 512, please explain what would be the impact of either mother or father who completed the questionnaire.

Reply: We thank the Reviewer for this comment. We acknowledge that having questionnaires completed by either mothers or fathers may have influenced the data. Evidence from the literature, such as studies on the Strengths and Difficulties Questionnaire (SDQ), indicates that fathers and mothers can differ in their ratings of child behavior, particularly for externalizing behaviors like hyperactivity. Fathers, for instance, tend to report higher levels of these behaviors than mothers, and interparental agreement is lower for abnormal behaviors compared to normal ones. This potential variability has been noted as a limitation in our study, although we believe it does not significantly affect the overall findings.

Changes: lines 515-519: Finally, with regard to the questionnaires that were meant to be completed by parents, some were filled out by mothers while others were completed by fathers, which may have introduced variability. Previous research has shown that fathers and mothers may differ in their reporting of certain behaviors, particularly for externalizing problems. [Davé, S., Nazareth, I., Senior, R., & Sherr, L. (2008). A comparison of father and mother report of child behaviour on the Strengths and Difficulties Questionnaire. Child psychiatry and human development, 39(4), 399–413. https://doi.org/10.1007/s10578-008-0097-6]

Reference #48 “Davé, S., Nazareth, I., Senior, R., & Sherr, L. (2008). A comparison of father and mother report of child behaviour on the Strengths and Difficulties Questionnaire. Child psychiatry and human development, 39(4), 399–413. https://doi.org/10.1007/s10578-008-0097-6” has been added at the bottom of the references list as well.

- In recommendation about further research, line 517, what is entail in the extensive neuropsychological and neuropsychiatric assessments within family clusters to be included in further study?

Reply: We suggest that a more comprehensive evaluation of family clusters would involve administering the full Leiter-3 battery, aiming for a more thorough assessment of cognitive performance in children. Analyzing patients with pre-pandemic IQ data would be highly valuable, but this seems nearly impossible when evaluating a cohort of healthy patients with no neuropsychiatric pre-existing conditions. Additionally, we recommend incorporating the Youth Self Report (YSR) for children as part of a broader evaluation. This would allow for a more nuanced comparison of emotional and behavioral problems, alongside the CBCL completed by their parents, to better understand the children's subjective perceptions. For a more rigorous evaluation, self-report questionnaires intended for parents should ideally be completed by both mothers and fathers. If this is not feasible, one parent should complete the questionnaire in a predetermined manner. The following text has been added in the discussion section.

Changes: lines 526-530: A more comprehensive evaluation should involve the full Leiter-3 battery to assess children's cognitive performance, alongside the Youth Self Report (YSR) and the CBCL for a nuanced comparison of emotional and behavioral issues, with self-report questionnaires completed by both parents when possible.

- Table 1, the row that has all zero value (Clinical classification: critical and MIS-C, post acute sequelae) can be omitted and put as narrative, also Figure 1, panel A. The graphic on CBCL for children aged 1.5-5 years contained no value, it could be omitted and just put as narrative.

Reply: We agree with the Reviewers, and we thank them for their suggestions.

Changes: Table 1 has been simplified as suggested, with the corresponding information now included in the text only (lines 259-265). Regarding Figure 1 - Panel A, we would like to retain the 1.5-5 year age class to ensure consistency with Panel B, which includes all the age classes analyzed. While children aged 1.5-5 years had scores below the clinical cut-off for total, internalizing, and externalizing problems subscales, this remains a relevant result worth reporting in the figure to facilitate comparisons with other age classes. As mentioned in the main text, “For all children aged 1.5-5 years, scores were below the clinical cut-off for total, internalizing, and externalizing problems subscales (Fig 1 and S5 Table) overall and stratified according to sociodemographic and COVID-19-related factors (S6-S7 Tables)” (lines 322-324).

---

## [Decision Letter · Decision Letter 2]

5 Mar 2025

Long-term neuropsychiatric and neuropsychological impact of the pandemic in Italian COVID-19 family clusters, including children and parents

PONE-D-23-36142R2

Dear Dr. Di Chiara,

We’re pleased to inform you that your manuscript has been judged scientifically suitable for publication and will be formally accepted for publication once it meets all outstanding technical requirements. I would like to sincerely apologise for the delay you have incurred with your submission. 

Kind regards,

Miquel Vall-llosera Camps

Senior Staff Editor

PLOS One

Reviewers' comments:

Reviewer's Responses to Questions

**Comments to the Author**

1. If the authors have adequately addressed your comments raised in a previous round of review and you feel that this manuscript is now acceptable for publication, you may indicate that here to bypass the “Comments to the Author” section, enter your conflict of interest statement in the “Confidential to Editor” section, and submit your "Accept" recommendation.

Reviewer #1: All comments have been addressed

Reviewer #2: All comments have been addressed

2. Is the manuscript technically sound, and do the data support the conclusions?

Reviewer #1: Yes

Reviewer #2: Yes

3. Has the statistical analysis been performed appropriately and rigorously? 

Reviewer #1: Yes

Reviewer #2: Yes

4. Have the authors made all data underlying the findings in their manuscript fully available?

Reviewer #1: Yes

Reviewer #2: Yes

5. Is the manuscript presented in an intelligible fashion and written in standard English?

Reviewer #1: Yes

Reviewer #2: Yes

6. Review Comments to the Author

Reviewer #1: (No Response)

Reviewer #2: Authors exhaustively addressed all the criticisms raised in the previous review and the paper is now suitable for publication.

In the present version there are no concerns about dual publication, research ethics or publication ethics.

7. PLOS authors have the option to publish the peer review history of their article (what does this mean? ). If published, this will include your full peer review and any attached files.

**Do you want your identity to be public for this peer review?** For information about this choice, including consent withdrawal, please see our Privacy Policy .

Reviewer #1: **Yes: ** Francesca Felicia Operto

Reviewer #2: **Yes: ** Giangennaro Coppola

---

## [Editor Report · Acceptance letter]

PONE-D-23-36142R2

PLOS ONE

Dear Dr. Di Chiara,

I'm pleased to inform you that your manuscript has been deemed suitable for publication in PLOS ONE. Congratulations! Your manuscript is now being handed over to our production team.

Kind regards,

on behalf of

Dr. Bao-Liang Zhong

Academic Editor

PLOS ONE